# The impact of long-term conditions on disability-free life expectancy: A systematic review

**Ilianna Lourida**, **Holly Q. Bennett**, **Fiona Beyer**, **Andrew Kingston**[‡]*,
**Carol Jagger**[‡]

Faculty of Medical Sciences, Population Health Sciences Institute, Newcastle University, Newcastle upon Tyne, United Kingdom

‡ These authors are joint senior authors on this work.
* andrew.kingston@newcastle.ac.uk

**Data Availability Statement:** All data are in the paper and its Supporting Information files.

**Funding:** This work was funded by the Dunhill Medical Trust (grant number RPGF1806\44 – CJ

## Abstract

Although leading causes of death are regularly reported, there is disagreement on which long-term conditions (LTCs) reduce disability-free life expectancy (DFLE) the most. We aimed to estimate increases in DFLE associated with elimination of a range of LTCs. This is a comprehensive systematic review and meta-analysis of studies assessing the effects of LTCs on health expectancy (HE). MEDLINE, Embase, HMIC, Science Citation Index, and Social Science Citation Index were systematically searched for studies published in English from July 2007 to July 2020 with updated searches from inception to April 8, 2021. LTCs considered included: arthritis, diabetes, cardiovascular disease including stroke and peripheral vascular disease, respiratory disease, visual and hearing impairment, dementia, cognitive impairment, depression, cancer, and comorbidity. Studies were included if they estimated HE outcomes (disability-free, active or healthy life expectancy) at age 50 or older for individuals with and without the LTC. Study selection and quality assessment were undertaken by teams of independent reviewers. Meta-analysis was feasible if three or more studies assessed the impact of the same LTC on the same HE at the same age using comparable methods, with narrative syntheses for the remaining studies. Studies reporting Years of Life Lost (YLL), Years of Life with Disability (YLD) and Disability Adjusted Life Years (DALYs = YLL+YLD) were included but reported separately as incomparable with other HE outcomes (PROSPERO registration: CRD42020196049). Searches returned 6072 unique records, yielding 404 eligible for full text retrieval from which 30 DFLE-related and 7 DALY-related were eligible for inclusion. Thirteen studies reported a single condition, and 17 studies reported on more than one condition (two to nine LTCs). Only seven studies examined the impact of comorbidities. Random effects meta-analyses were feasible for a subgroup of studies examining diabetes (four studies) or respiratory diseases (three studies) on DFLE. From pooled results, individuals at age 65 without diabetes gain on average 2.28 years disability-free compared to those with diabetes (95% CI: 0.57–3.99, p<0.01, $I^2$ = 96.7%), whilst individuals without respiratory diseases gain on average 1.47 years compared to those with respiratory diseases (95% CI: 0.77–2.17, p<0.01, $I^2$ = 79.8%). Eliminating diabetes, stroke, hypertension or arthritis would result in compression of disability. Of

and AK). The funding source had no role in the study design, data collection, data analysis, data interpretation, writing of the manuscript or decision to submit the paper for publication.

**Competing interests:** The authors have declared that no competing interests exist.

the seven longitudinal studies assessing the impact of multiple LTCs, three found that stroke had the greatest effect on DFLE for both genders. This study is the first to systematically quantify the impact of LTCs on both HE and LE at a global level, to assess potential compression of disability. Diabetes, stroke, hypertension and arthritis had a greater effect on DFLE than LE and so elimination would result in compression of disability. Guidelines for reporting HE outcomes would assist data synthesis in the future, which would in turn aid public health policy.

## Introduction

Chronic or long-term conditions (LTCs) affect the health and quality of life of millions of people globally and exert heavy pressure on health care services. A long-term condition is a condition that cannot, at present, be cured but is controlled by medication and/or other treatment/ therapies. It is estimated that nearly 40% of the UK and US populations have at least one LTC [1], while one in three older adults live with multiple LTCs, and these figures are projected to rise dramatically by 2035 [2]. LTCs including heart disease, diabetes, cancer, and respiratory diseases have been among the leading causes of death globally in recent decades although mortality rates have declined due to improved medical care and availability [3]. However, the increased longevity coupled with the rising prevalence of LTCs has led to concerns about the impact on disability burden especially among older people. Therefore, the focus is gradually shifting from mortality and life expectancy (LE) as the measure of success to the need to improve quality of life and using disability-free life expectancy (DFLE; the number of years from a particular age spent free of disability) or healthy life expectancy (HLE; the number of years from a particular age spent in good health) (See S1 Methods for full definitions).

Previous research has shown that individual LTCs such as diabetes [4,5], arthritis [6], depression [7,8], and sensory impairments [9] have a significant effect on DFLE. Other studies have attempted to break down the disability burden of multiple LTCs and have identified heart disease [10–12], stroke [8,13] or dementia [14] as the most important causes of DFLE and/or LE loss. In addition, a few studies have examined the issue of comorbidity trying to assess whether disability could be attributed to one or more LTCs [8,10,13,15]. The potential changes in DFLE and LE resulting from the elimination of various LTCs has also been estimated in some populations [12,16–18] to determine whether better control of these conditions would postpone the onset of disability. The gains in disability-free and total life expectancy can vary depending on the age at which estimates are reported, and especially the condition that is hypothetically eliminated.

The Global Burden of Disease studies provide country-specific and global assessments of Years of Life Lost (YLL), Years of Life with Disability (YLD) and Disability Adjusted Life Years (DALYs = YLL+YLD). However, DALYs are typically reported for all ages, and there is currently no consensus on which LTCs are the main contributors to reductions in DFLE at older ages, or which LTCs if eliminated will result in greater gains in DFLE than LE driving compression of disability. This is important to inform health policy and priority setting in terms of prevention and treatment of LTCs, design of appropriate care packages, and for identifying research and funding priorities for specific conditions. Despite the growing number of studies assessing the impact of common LTCs on health expectancy outcomes, no systematic review has synthesised the available evidence.

Our aim was to address this gap by conducting a systematic review of the literature to assess the effect of a range of LTCs, singly and in combination, on disability-free and total life

expectancy, and specifically, which LTCs have a greater effect on DFLE than LE. A fuller description of these key concepts is provided in S1 Methods.

## Methods

### Data sources and searches

The systematic review was conducted following PRISMA guidelines (S1 Checklist), and the general principles published by the NHS Centre for Reviews and Dissemination (CRD) [19,20]. A protocol was developed following discussion with methods and topic experts and is registered with PROSPERO (PROSPERO 2020: CRD42020196049).

 The search strategy was developed by an experienced information specialist (FB) in collaboration with the review team based on the concepts [long-term conditions] AND [disability-free life expectancy]. The search was developed for MEDLINE (OVID, see S2 Methods) using thesaurus headings, and title, abstract and keyword field terms adapted as appropriate for the other electronic databases (MEDLINE and In-Process & Other Non-indexed citations, Embase [via OVID], HMIC Health Management Information Consortium, Science Citation Index, Social Science Citation Index [via Web of Science]). Searches were initially limited to studies published in English from July 2007 to July 2020, 2007 being selected as the date of the publication of the first paper reporting the effect of LTCs on disability-free life expectancy [41]. Update searches were made as comprehensive as possible and conducted from inception to 8 April 2021. Studies were excluded by record type in MEDLINE and Embase if they were editorials, opinion pieces or letters. Reference lists of all included articles were checked for additional relevant studies.

### Eligibility criteria

The aim of the review was to assess the effect of LTCs, singly and in combination, on disability-free (DFLE) and total life expectancy (LE), and specifically, whether LTCs had a greater effect on DFLE or LE. If the former, then elimination of the LTC could result in a compression of disability. To ensure that no studies on DFLE were missed, we included wider health expectancy terms of health-adjusted LE (HALE) and Healthy Life Expectancy (HLE). LTCs included non-communicable chronic, long-term or life-limiting conditions or diseases. Relevant LTCs were identified through previous research [21,22]: arthritis, diabetes, cardiovascular disease including stroke and peripheral vascular disease, respiratory disease, visual and hearing impairment, dementia, cognitive impairment, depression, cancer, and comorbidity/multimorbidity. For this review, obesity was considered a risk factor for disease rather than a chronic condition or disease. Therefore, studies reporting obesity as a single exposure were not eligible. Studies were included if they measured the effect of any of the above LTCs on health expectancy outcomes, primarily disability-free, active or healthy life expectancy reported at age 50 and older compared to individuals without the LTC. Studies reporting estimates at birth only were excluded. Outcomes reported as gains in DFLE or HALE with and without a specific LTC (or multimorbidity) and estimates from projections of DFLE were also eligible. Total life expectancy was eligible as an outcome only when reported alongside the main outcomes (DFLE, ALE, HALE).

### Study selection

Search results were downloaded to Endnote version X9 (Clarivate, Philadelphia, PA) and de-duplicated before being imported to the Rayyan reference management software [23] for screening. Starting with a random sample of 100 citations, titles and abstracts were screened

for relevance independently by pairs of reviewers (IL and HB, AK or CJ). Following clarification of concepts and criteria, the team completed the screening process with randomly assigned citations for each pair of reviewers. Disagreements were resolved by discussion adjudicated by a third reviewer if necessary. The full text of relevant articles was retrieved and screened in the same way using the predefined eligibility criteria.

## Data extraction

Two reviewers piloted and refined a data extraction form using two randomly selected studies (IL and CJ). Data for the remaining studies were extracted independently by pairs of reviewers in a structured form using Excel. Publication details (first author, year, country), study and population characteristics (study design, years of analysis and number of follow-ups where relevant, sample size, age, % men/women), LTCs, outcomes and their measurement, brief methods of analysis, age at which outcomes were reported (prioritising age 65 where possible), and quantitative results for LE and health expectancies with and without the LTC were recorded. Discrepancies were resolved by discussion and involvement of a third reviewer (CJ) where necessary. Authors of four papers were contacted for clarification or additional information.

Searches retrieved papers from the GBD group reporting Years of Life Lost (YLL), Years of Life with Disability (YLD) and Disability Adjusted Life Years (DALYs = YLL+YLD). These studies were screened separately, and grouped into global, regional, and national estimates. We focused on global estimates reporting estimates at more than one time point and excluded any studies that did not provide estimates of both YLL and YLD for individual conditions. However, estimates of YLD were reported for all ages, or age-adjusted, and thus were not directly comparable with DFLE/HALE estimates at older ages. We therefore use these studies to compare the relative ranking of our selected LTCs, and how these have changed over time.

## Quality assessment

The quality of included studies was assessed by one reviewer and checked by a second. In an amendment to the protocol, a modified checklist was used combining items from the Joanna Briggs Institute (JBI) critical appraisal tools for prevalence and longitudinal studies [24,25] with criteria previously used by Freedman [26] for evaluating surveys of trends in self-reported disability. Full details are given in S3 Methods.

## Data synthesis and analysis

Studies were categorised by study design, LTC, and health expectancy outcome, and summaries of these characteristics were provided in tabular and narrative form. Meta-analyses to estimate summary measures of the impact of the LTCs on health expectancy at a specific age were considered to be feasible for three or more studies assessing the impact of the same LTC on the same health expectancy outcome (e.g., DFLE with disability measured by performance in ADLs) at the same age (e.g., at 65) using comparable methods. The mean difference and 95% confidence intervals in DFLE/HALE (years) between those without and those with the LTC were calculated providing sufficient information was reported in the included studies. Data were pooled using random-effects models and presented in forest plots separately for men and women. Statistical heterogeneity was explored through the $I^2$ and the Q tests according to specific categories (low = 25%, moderate = 50%, and high = 75%). Funnel plots were used to evaluate potential publication bias. Statistical analyses were conducted using Stata (StataCorp 2019 Stata Statistical Software: Release 16). Studies that could not be included in meta-analyses due to important differences in key characteristics (e.g., study design, age at which estimates were reported, outcome, data availability) were synthesised narratively.

## Results

### Study selection

The electronic searches yielded 6,072 unique citations. Screening of titles and abstracts against the eligibility criteria resulted in the retrieval of the full text of 404 articles of which 29 DFLE-related and seven DALY-related were eligible for inclusion. One additional article was identified through reference list checking of the included studies. In total, 37 articles met the inclusion criteria (Fig 1).

### Study characteristics

The characteristics of the included DFLE-related studies are presented in Table 1. One of the identified studies was an independent report [27] and the rest were peer-reviewed publications

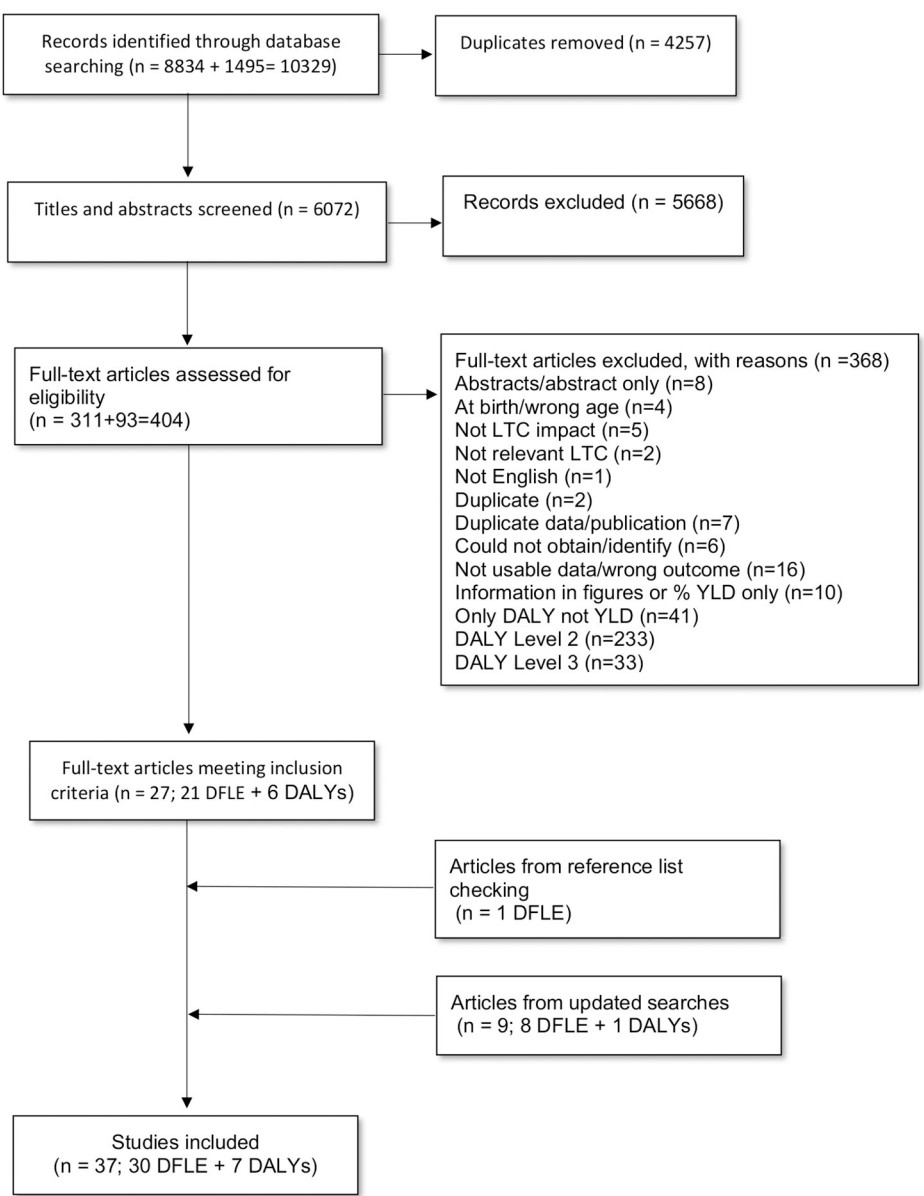

**Fig 1. Study selection process for the systematic review.**

**Table 1. Characteristics of included studies examining the impact of long-term conditions (LTCs) on disability-free life expectancy.**

| Study (year) | Survey | LTCs measured | Survey years (n of measurements) | Population (age, sample size) | Outcomes (disability measurement) | Method used to calculate DFLE (or HALE) |
|---|---|---|---|---|---|---|
| **Cross-sectional studies** | | | | | | |
| Bronnum-Hansen (2006) [12] | Danish Health Interview Survey 2000 | Chronic obstructive lung disease, Diabetes, Cerebrovascular disease, Ischaemic heart diseases, Neoplasms | 1995–1999 (1) | ≥65 years; 3,009 ppts | LE, Expected lifetime without long-standing illness | Sullivan, Cause elimination |
| Campolina (2013) [16] | Saude, Bem-Estare Envelhecimento (SABE) [Health, Wellbeing and Ageing] study (Brazil) | Cancer, Cerebrovascular disease, Diabetes, Heart disease, (systemic arterial) Hypertension, Lung disease | 2000 (1) | ≥60 years; 2,143 ppts | Gain in LE, DFLE (ADLs) | Sullivan, Cause elimination |
| Campolina (2014) [17] | Saude, Bem-Estare Envelhecimento (SABE) [Health, Wellbeing and Ageing] study (Brazil) | Cancer, Cerebrovascular disease, Diabetes, Heart disease, (systemic arterial) Hypertension, Lung disease | 2010 (1) | ≥60 years; 907 ppts | Gain in LE, DFLE (ADLs) | Sullivan, Cause elimination |
| Chen (2014) [39] | The 2006 China Disability Survey | Cerebrovascular disease, Hearing loss, Osteoarthritis | 2006 (1) | ≥60 years; 354,859 ppts | Life expectancy with disability (LED) (abnormalities in anatomical structure or loss of a certain organ or function (either psychological or physiological), and lost ability to perform an activity in normal way) | Sullivan |
| Hu (2019) [29] | Global Burden of Disease Study 2016 for China | Cancer, Cardiovascular diseases, Chronic respiratory diseases, Diabetes | 1990, 2016 (1) | ≥50 years; unclear | Gain in DFLE (Global burden of disease YLD) | Sullivan, Cause elimination |
| Huo (2016) [34] | Australian Survey of Disability, Ageing and Carers, and Australian Diabetes, Obesity and Lifestyle study | Diabetes | 2011–2012 (1) | ≥50 years; unclear | Gain in LE, DFLE (ADLs, mobility, communication) | Chiang, Sullivan, Cause elimination |
| Manton (1991) [35] | National Long Term Care Surveys (USA) | Dementia | 1983–1984 (1) | ≥65 years; (unlear) | Gain in ALE (ADLs) | Cause elimination |
| Murtaugh (2011) [14] | National Mortality Followback Survey (USA) | Arthritis, COPD, Dementia, Diabetes, Heart attack, Stroke | 1993 (1) | Unclear | LE, DLE (physical strength & endurance, ADLs, IADLs, mobility) | Projected lifetime risk |
| Nusselder (1996) [18] | Dutch National Survey of General Practice | Cancer, Chronic non-specific lung disease, Diabetes, Heart disease | 1987–1988 (1) | ≥15 years; 10,147 ppts | Gain in LE, DFLE (OECD indicator) | Sullivan, Cause elimination |
| Mathers (1999) [11] | Survey of disability, ageing and carers (Australia) | Cerebrovascular disease, COPD, Diabetes, Hearing loss, Hypertensive disease Ischaemic heart disease, Osteoarthritis, Rheumatoid arthritis | 1993 (1) | ≥5 years; unclear | Gain in LE, HALE (range of disabilities, impairments and handicap) | Cause elimination |
| Public Health Agency of Canada report (2012) [27] | Canadian Community Health Survey | Diabetes, Hypertension, + comorbidity | 2000–2005 (1) | ≥20 years; 200,809 (diabetes), 173,567 (hypertension) ppts | Gains in LE, HALE (physiological or psychological functioning measured by HUI3) | Sullivan, cause elimination |

(*Continued*)

**Table 1.** (Continued)

| Study (year) | Survey | LTCs measured | Survey years (n of measurements) | Population (age, sample size) | Outcomes (disability measurement) | Method used to calculate DFLE (or HALE) |
|---|---|---|---|---|---|---|
| | Canadian Community Health Survey | Cancer | 2002–2005 (1) | >0 years; 156,020 ppts | LE, HALE (physiological or psychological functioning measured by HUI3) | Sullivan, Cause elimination |
| Sikdar (2010) [36] | Canadian Community Health Survey (restricted to Newfoundland and Labrador residents) | Diabetes | 2001–2005 (1) | ≥15 years; 3,567 ppts | Gains in LE, HALE (physiological or psychological functioning measured by HUI3) | Sullivan, Cause elimination |
| Steensma (2016) [37] | Canadian Community Health Survey | Depression | 2009–2010 (1) | ≥20 years; 103,815 ppts | Period LE, HALE (physiological or psychological functioning measured by HUI3) | Sullivan |
| **Longitudinal studies** | | | | | | |
| Andrade (2010) [4] | Mexican Health and Aging Study | Diabetes | 2001–2003 (2) | ≥50 years; 11,929 for ADLs, 11,944 for IADLs, 11,935 for Nagi | LE, DFLE (ADLs, IADLs & Nagi physical performance limitations) | Multi-state life tables |
| Bardenheier (2016a) [5] | Health and Retirement Study | Diabetes | Cohort I: 1992–2002 (6) Cohort II: 2002–2012 (6) | 50–70 years; Cohort I: 9,754, Cohort II: 3,027 ppts | LE, DFLE (ADLs, IADLs & mobility) | Multi-state life tables |
| Bardenheier (2016b) [31] | Health and Retirement Study | Diabetes | 1998–2012 (8) | ≥50 years; 20,008 ppts | LE, DFLE (ADLs, IADLs & mobility) | Multi-state life tables |
| Belanger (2002) [38] | National Population Health Survey (Canada) | Arthritis, Cancer, Diabetes | 1994–1996 (2) | ≥45 years; 8,009 ppts | LE, DFLE (activity limitations, dependency) | Multi-state life tables |
| Chiu (2019) [32] | Nihon University Japanese Longitudinal Study of Aging | Stroke | 1999–2009 (5) | ≥65 years; 4,833 ppts | LE, DFLE (ADLs, IADLs) | Multi-state life tables |
| Diehr (1998) [30] | Cardiovascular Health Study (USA) | Cardiovascular disease | Unclear | ≥65 years; 5,201 ppts | HLE (self-reported health status) | Transition probabilities |
| Dodge (2003) [33] | MoVIES survey (USA) | Alzheimer's disease | 1989–1995 (3?) | ≥70 years; 1,201 ppts | LE, DLE (IADLs) | Multi-state life tables |
| Fang (2009) [28] | Beijing Multidimensional Longitudinal study on Aging (China) | Stroke | Cohort I: 1992–1997 (2) Cohort II: 2000–2004 (2) | ≥55 years; Cohort I: 3,227, Cohort II: 2,837 ppts | LE, ALE (WHO disability scale) | Multi-state life tables |
| Hayward (1998) [40] | Longitudinal Study of Aging (USA) | Heart disease, cerebrovascular disease, cancer, + comorbidity | 1984–1990 (3) | ≥70 years; 7,527 ppts | LE, ALE (ADLs, IADLs) | Multi-state life tables, cause elimination |
| Jagger (2003) [10] | GP health assessments (UK) | Diabetes | 1990–1999 (5) | ≥75 years; 2,474 ppts | LE, ALE (ADLs) | Multi-state life tables |
| Jagger (2007) [41] | MRC Cognitive Function and Ageing Study (UK) | Arthritis, Coronary heart disease, Chronic airway obstruction, Diabetes, Peripheral vascular disease, Stroke, Visual impairment, Hearing impairment, Cognitive impairment, + comorbidity (arthritis and comorbidity, CHD and comorbidity) | 1992–2002 (4) | ≥65 years; 12,881 ppts | LE, DFLE (ADLs, IADLs) | Multi-state life tables |

(*Continued*)

**Table 1.** (Continued)

| Study (year) | Survey | LTCs measured | Survey years (n of measurements) | Population (age, sample size) | Outcomes (disability measurement) | Method used to calculate DFLE (or HALE) |
|---|---|---|---|---|---|---|
| Laditka (2016) [13] | Panel Study of Income Dynamics (USA) | Arthritis, Depression, Diabetes, Heart disease, Hypertension, Lung disease, Memory, Stroke, + comorbidity (diabetes and combinations of the other LTCs) | 1999–2011 (7) | ≥55 years; 2,118 ppts | LE, DFLE (ADLs) | Multi-state life tables |
| Liang (2020) [15] | Taiwan Longitudinal Study on Aging | Diabetes, Hypertension, + comorbidity | 1996–2011 (4) | ≥50 years; 5,131 ppts | LE, DFLE (ADLs) | Multi-state life tables |
| Pérès (2008) [7] | MRC Cognitive Function and Ageing Study (UK) | Depression, Emotional problems, + Multimorbidity | 1998–2008 (4) | ≥65 years; 11,022 ppts | LE, DFLE (ADLs, IADLs) | Multi-state life tables |
| Reynolds (2008a) [6] | Asset and Health Dynamics Among the Oldest Old-AHEAD (USA) | Arthritis | 1993–1998 (3) | ≥70 years; 7,381 ppts | LE, ALE (ADLs) | Multi-state life tables |
| Reynolds (2008b) [8] | Asset and Health Dynamics Among the Oldest Old-AHEAD (USA) | Depression, Cancer, Diabetes, Heart disease, Stroke, + comorbidity (depression and each of the other LTCs) | 1993–1998 (3) | ≥70 years; 7,381 ppts | LE, ALE (ADLs) | Multi-state life tables |
| Tareque (2019) [9] | Panel on Health and Ageing in Singaporean Elderly | Visual impairment, Hearing impairment, + comorbidity | 2009–2015 (3) | ≥60 years; 3,452 ppts | LE, DFLE (physical function limitations & ADLs, IADLs) | Multi-state life tables |

ADLs: Activities of daily living, ALE: Active life expectancy, DFLE: Disability-free life expectancy, DLE: Disabled life expectancy, HALE: Health-adjusted life expectancy, HLE: Healthy life years, HUI3: Health Utilities Index Mark 3 instrument, IADLs: Instrumental activities of daily living, LE: Life expectancy, LTCs: Long-term conditions, OECD: Organisation for Economic Cooperation and Development, WHO: World Health Organisation.

spanning from 1991 to 2020. Thirteen studies were cross-sectional and 17 longitudinal, with data from the USA (n = 10), Canada (n = 4), UK (n = 3), Australia (n = 2), Brazil (n = 2), China (n = 3); and Mexico, Japan, Taiwan, Singapore, Denmark, and the Netherlands (one study each). Three studies reported the impact of LTCs on DFLE at more than one time point [5,28,29]. Most studies reported outcomes for populations aged 60 years or older, and over half of studies (63%) reported estimates at more than one age. Gender distribution varied, with all but three studies [10,28,30] presenting results separately for men and women, and a fourth study stratifying results by sex and race/ethnicity [13]. The number of assessments in longitudinal studies ranged from two to eight with a maximum length of follow-up of 15 years. Most studies reported using nationally representative data for non-institutionalised populations, with self-reported diagnosis of the LTCs coded using standardised criteria- mostly the International Classification of Diseases. Details of the quality assessment items and ratings are shown in S1 Results and S1 and S2 Tables.

## Overview of LTCs and outcomes

The LTCs specified in the inclusion criteria and identified in the included studies were mainly self-reported. Thirteen studies reported a single condition [4–6,10,28,30–37], while the remaining 17 [7–18,27,29,38–40] reported the impact of more than one condition ranging from two to nine LTCs. Despite the high number of studies reporting more than one LTC,

only seven [8,9,13,15,27,40,41] examined the impact of specified multiple LTCs on DFLE or HALE.

Twenty-five studies reported disability-related outcomes (DFLE, ALE, DLE), four HALE [11,27,30,36], and for one [12], the outcome was expected lifetime without long-standing illness. Disability was mostly measured as limitations in ADLs or in combination with IADLs and/or other functions (e.g., physical strength and endurance, mobility, communication). Health-related quality of life was typically measured using the Health Utilities Index Mark instrument towards the estimation of HALE or self-rated health status (excellent, very good, good, fair, poor).

### Effect of LTCs on health expectancy

We report the two LTCs for which meta-analysis was deemed feasible (diabetes, respiratory diseases). We report in brief the remaining conditions (cardiovascular disease, hypertension, cerebrovascular disease, cancer, arthritis, sensory loss, dementia and cognitive impairment, depression) with more detail provided in S2 Results.

### Diabetes

Nineteen studies [4,5,8,10–18,27,29,31,34,36,38,41] (Table 2) examined the impact of diabetes on health expectancy. Four out of the 19 studies (representing five population samples) provided estimates of SEs or 95% CI of DFLE at age 65 and could be included in a meta-analysis. The meta-analysis showed individuals at age 65 without diabetes gain on average 2 years disability-free compared to those with diabetes (pooled overall mean difference in DFLE = 2.28 years, 95% CI: 0.57–3.99, p<0.01, $I^2$ = 96.7%). Women gained a slightly higher number of years (pooled mean difference in DFLE = 2.51 years, 95% CI: -0.28–5.31, p<0.01, $I^2$ = 96.6%) than men (pooled mean difference in DFLE = 2.06 years, 95% CI: -0.24–4.36, p<0.01, $I^2$ = 96.4%; Fig 2). The funnel plot indicated potential publication bias (S1 Fig). Results of the Egger test for small-study effects suggested this is unlikely to be problematic (p = 0.10). Meta-analysis of LE estimates for the above four studies was not possible due to insufficient data.

Six additional studies [10,11,14,15,18,36] evaluated the effect of diabetes at age 65. Of these, three [11,18,36] focussed on elimination of diabetes, reporting small gains in LE (range of gain in LE: 0.10 to 1.8 years), and in two studies [18,36], the gain in DFLE years was greater than that in LE signifying a compression of disability after diabetes elimination. Results of the nine studies reporting DFLE at other ages suggest a similar pattern with stronger impact reported for younger ages and some variation by gender (details provided in S2 Results). Most of the studies therefore concluded that elimination of diabetes would result in a compression of disability.

### Respiratory diseases

Nine studies examined the impact of respiratory diseases (including COPD, bronchitis, emphysema, asthma) on health expectancy (Table 3), of which three -representing four population samples- were included in a meta-analysis. Pooled results indicated that at age 65 individuals without respiratory diseases gain on average 1.5 years disability-free compared to those with respiratory diseases (pooled overall mean difference in DFLE = 1.47 years, 95% CI: 0.77–2.17, p<0.01, $I^2$ = 79.8%). Men gained a similar number of years (pooled mean difference in DFLE = 1.40 years, 95% CI: 0.50–2.30, p<0.01, $I^2$ = 77.7%) to women (pooled mean difference in DFLE = 1.54 years, 95% CI: 0.35–2.73, p<0.01, $I^2$ = 82.1%; Fig 3). The funnel plot indicated potential publication bias (S2 Fig) but results of Egger's test for small-study effects suggested

**Table 2. Summary data for the impact of diabetes on life expectancy and disability-free life expectancy (or health-adjusted life expectancy) for men and women.**

| Study (year) | Survey | At age | LTC measured | Difference in LE Mean (95% CI) | | Difference in DFLE (or HALE) Mean (95% CI) | |
|---|---|---|---|---|---|---|---|
| | | | | Men | Women | Men | Women |
| *Belanger (2002) [38] | National Population Health Survey (Canada) | 45 years | Diabetes (vs. no diabetes) | 5.6 | 13.0 | 10.7 | 14.1 |
| *Laditka (2016) [13] | Panel Study of Income Dynamics (USA) | 55 years | Diabetes (vs. no diabetes) | AA: 4.5 W: 6.2 | AA: 7.3 W: 13.5 | AA: 7.8 W: 9.5 | AA: 10.4 W: 14.2 |
| *Public Health Agency of Canada report (2012) [27] | Canadian Community Health Survey | 55 years | Diabetes (vs. no diabetes) | Gain†: 1.2 | Gain: 1.4 | Gain: 1.2 | Gain: 1.2 |
| Andrade (2010) [4] | Mexican Health and Aging Study | 60 years | Diabetes (vs. no diabetes) | 9.8 | 7.1 | 8.9 | 6.2 |
| Bardenheier (2016a) [5] | Health and Retirement Study | 60 years | Diabetes (vs. no diabetes) | Cohort 1: 0.9 Cohort 2: 0.8 | Cohort 1: 0.7 Cohort 2: 0.5 | Cohort 1: 1.9 Cohort 2: 1.2 | Cohort 1: 1.9 Cohort 2: 1.2 |
| Bardenheier (2016b) [31] | Health and Retirement Study | 60 years | Diabetes (vs. no diabetes) | 4.0 | 4.1 | 4.3 | 4.9 |
| *Campolina (2013) [16] | SABE study (Brazil) | 60 years | Diabetes (vs. no diabetes) | Gain: 7.0 | Gain: 2.3 | Gain: 6.5 | Gain: 8.3 |
| *Campolina (2014) [17] | SABE study (Brazil) | 60 years | Diabetes (vs. no diabetes) | Gain: 2.8 | Gain: 1.9 | Gain: 5.7 | Gain: 11.1 |
| *Bronnum-Hansen (2006) [12] | Danish Health Interview Survey 2000 | 65 years | Diabetes (vs. no diabetes) | 0.1 | 0.1 | 0.2 (-0.30 to 0.70) | 0.2 (-0.40 to 0.80) |
| *Hu (2019) [29] | Global Burden of Disease Study 2016 for China | 65 years | Diabetes (vs. no diabetes) | .. | .. | Cohort 1: 0.14 (-0.74 to 1.02) Cohort 2: 0.33 (-0.93 to 1.59) | Cohort 1: 0.22 (-0.95 to 1.39) Cohort 2: 0.32 (-1.38 to 2.02) |
| Huo (2016) [34] | Australian Survey of Disability, Ageing and Carers, and Australian Diabetes, Obesity and Lifestyle study | 65 years | Diabetes (vs. no diabetes) | Gain: 1.4 (1.3 to 1.6) | Gain: 1.7 (1.5 to 1.9) | Gain: 5.8 (4.2 to 7.3) | Gain: 6.8 (5.6 to 8.0) |
| Jagger (2003) [10] | GP health assessments (UK) | 65 years | Diabetes (vs. no diabetes) | 4.7 | | | 3.8 |
| *Jagger (2007) [41] | MRC Cognitive Function and Ageing Study (UK) | 65 years | Diabetes (vs. no diabetes) | 4.4 (3.2 to 5.6) | 5.6 (4.3 to 6.9) | 4.1 (2.8 to 5.4) | 5.1 (3.4 to 6.8) |
| *Liang (2020) [15] | Taiwan Longitudinal Study on Aging | 65 years | Diabetes (vs. no diabetes) | 4.2 | 4.0 | 4.5 | 4.2 |
| *Mathers (1999) [11] | Survey of disability, ageing and carers (Australia) | 65 years | Diabetes (vs. no diabetes) | Gain: 0.16 | Gain: 0.10 | Gain: 0.15 | Gain: 0.18 |
| *Murtaugh (2011) [14] | National Mortality Followback Survey (USA) | 65 years | Diabetes (vs. no diabetes) | -0.7 | 2.1 | DLE: 0.0 | DLE: 0.0 |
| *Nusselder (1996) [18] | Dutch National Survey of General Practice | 65 years | Diabetes (vs. no diabetes) | Gain: 0.3 | Gain: 0.1 | Gain: 0.5 | Gain: 0.2 |

*(Continued)*

**Table 2.** (Continued)

| Study (year) | Survey | At age | LTC measured | Difference in LE Mean (95% CI) | | Difference in DFLE (or HALE) Mean (95% CI) | |
|---|---|---|---|---|---|---|---|
| | | | | Men | Women | Men | Women |
| **Sikdar (2010) [36]** | Canadian Community Health Survey (restricted to Newfoundland and Labrador residents) | 65 years | Diabetes (vs. no diabetes) | Gain: 1.2 | Gain: 1.8 | Gain: 1.1 | Gain: 1.5 |
| *Reynolds (2008b) [8] | Asset and Health Dynamics Among the Oldest Old-AHEAD (USA) | 70 years | Diabetes (vs. no diabetes) | 4.8 (3.6 to 6.0) | 5.9 (4.7 to 7.1) | 4.1 (3.2 to 4.9) | 5.7 (4.8 to 6.6) |

*study measured multiple LTCs

†: Gain in LE/DFLE/HALE after elimination of LTC.

Difference in LE/DFLE: Years without LTC minus years with LTC, AA: African American, DFLE: Disability-free life expectancy, DLE: Life expectancy with disability, HALE: Health-adjusted life expectancy, LTC: Long-term condition, W: White.

bias is not problematic (p = 0.56). Meta-analysis of LE estimates for the above three studies was not possible due to insufficient data.

The remaining six studies [11,13,14,16–18] examining the impact of respiratory diseases reported estimates at 55, 60, and 65 years. Comparison of the difference in LE and DFLE between those without and with the LTC was available for five studies, and indicated that respiratory diseases had a greater impact on DFLE than LE in women in three of the studies [13,16,17] analysing mostly post-2000 cohort data (LE gain, range: 1.85–6.5 years; DFLE gain, range: 7.6–13.5 years). No substantial difference was observed for women in the other two studies [11,14], whereas results for men were mixed.

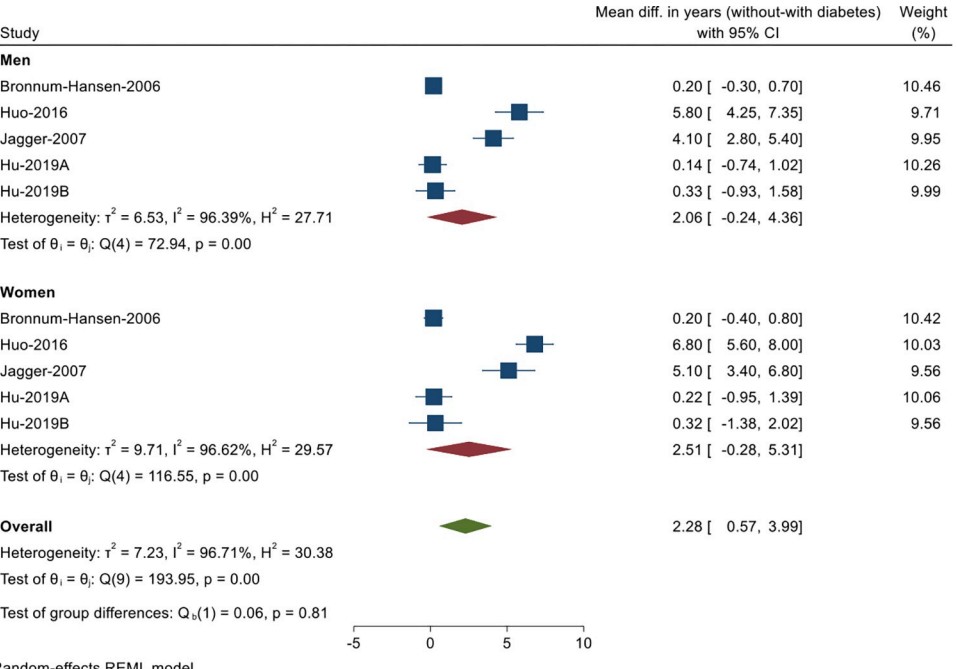

**Fig 2. Meta-analysis of mean difference in disability-free life expectancy (years) of individuals without diabetes compared to those with diabetes (based on data from 4 out of 19 studies reporting the impact of diabetes on disability-free life expectancy; Hu et al 2019 reported estimates for two population samples, A = in 1990, B = in 2016).**

**Table 3. Summary data for the impact of respiratory diseases on life expectancy and disability-free life expectancy (or health-adjusted life expectancy) for men and women.**

| Study (year) | Survey | At age | LTC measured | Difference in LE Mean (95% CI) | | Difference in DFLE (or HALE) Mean (95% CI) | |
|---|---|---|---|---|---|---|---|
| | | | | Men | Women | Men | Women |
| *Laditka (2016) [13] | Panel Study of Income Dynamics (USA) | 55 years | Lung disease (vs. no LTC) | AA: 1.8 W: 8.0 | AA: 6.5 W: 9.9 | AA: 7.8 W: 15.8 | AA: 12.7 W: 13.5 |
| *Campolina (2013) [16] | SABE study (Brazil) | 60 years | Lung disease (vs. no LTC) | Gain†: 7.13 | Gain: 2.16 | Gain: 6.70 | Gain: 7.61 |
| *Campolina (2014) [17] | SABE study (Brazil) | 60 years | Lung disease (vs. no LTC) | Gain: 2.8 | Gain: 1.85 | Gain: 5.85 | Gain: 10.68 |
| *Bronnum-Hansen (2006) [12] | Danish Health Interview Survey 2000 | 65 years | COPD (vs. no COPD) | Gain: 0.5 | Gain: 0.5 | Gain: 0.40 (-0.10 to 0.90) | Gain: 0.30 (-0.34 to 0.94) |
| *Hu (2019) [29] | Global Burden of Disease Study 2016 for China | 65 years | Chronic respiratory diseases (vs. no LTC) | .. | .. | Cohort 1: 1.82 (0.85 to 2.79) Cohort 2: 1.26 (-0.06 to 2.58) | Cohort 1: 1.91 (0.63 to 3.19) Cohort 2: 1.13 (-0.65 to 2.90) |
| *Jagger (2007) [41] | MRC Cognitive Function and Ageing Study (UK) | 65 years | Chronic airway obstruction (vs. no LTC) | Gain: 2.7 (1.9 to 3.4) | Gain: 2.2 (1.5 to 2.9) | Gain: 2.3 (1.45 to 3.15) | Gain: 2.8 (2.0 to 3.60) |
| *Mathers (1999) [11] | Survey of disability, ageing and carers (Australia) | 65 years | COPD (vs. no COPD) | Gain: 0.54 | Gain: 0.51 | Gain in HALE: 0.18 | Gain in HALE: 0.19 |
| *Murtaugh (2011) [14] | National Mortality Followback Survey (USA) | 65 years | COPD (vs. no COPD) | 0.9 | 3.9 | DLE: 0.1 | DLE: 0.0 |
| *Nusselder (1996) [18] | Dutch National Survey of General Practice | 65 years | Chronic non-specific lung disease (vs. no LTC) | Gain: 0.3 | Gain: 0.1 | Gain: 0.5 | Gain: 0.2 |

*study measured multiple LTCs

†: Gain in LE/DFLE/HALE after elimination of LTC.

Difference in LE/DFLE: Years without LTC minus years with LTC, AA: African American, DFLE: Disability-free life expectancy, DLE: Life expectancy with disability, HALE: Health-adjusted life expectancy, LTC: Long-term condition, W: White.

## Other health conditions

Twelve studies (seven cross-sectional and five longitudinal) assessed the impact of cardiovascular diseases on health expectancy at different ages [8,10–14,16–18,29,30,40] (Table 4). These studies show mixed evidence as to whether eliminating cardiovascular disease would result in compression or expansion of disability, with variation by age and gender. Four cross-sectional [11,16,17,27] and two longitudinal [13,15] studies assessed the effect of hypertension on health expectancy at various ages (Table 5). All but one [15] suggested that elimination of hypertension would lead to substantial gains in LE and DFLE resulting in a compression of disability. Six studies [11,12,16,17,39,40] evaluated the impact of cerebrovascular diseases on health expectancy (Table 6) with a further six on stroke specifically [8,10,13,14,28,32] (Table 7). While there was no clear evidence for compression of disability from eliminating cerebrovascular disease (Table 6), there was for stroke (Table 7). Ten studies evaluated the impact of cancer (Table 8) [8,11,12,16–18,27,29,38,40]. Apart from one [17], all reported that cancer had a major impact on LE, and a lesser impact on DFLE reduction; elimination would therefore likely result in an expansion of disability. Seven studies [6,10,13,14,38] examined the effect of arthritis [11,39] on health expectancy (Table 9). Arthritis was associated with a small loss of LE, generally greater for women. All studies reported a greater effect on DFLE than LE, resulting in a compression of disability if eliminated. Four studies [9–11,39] reported the impact of hearing impairment of which two [9,10] also included the impact of visual impairment on LE

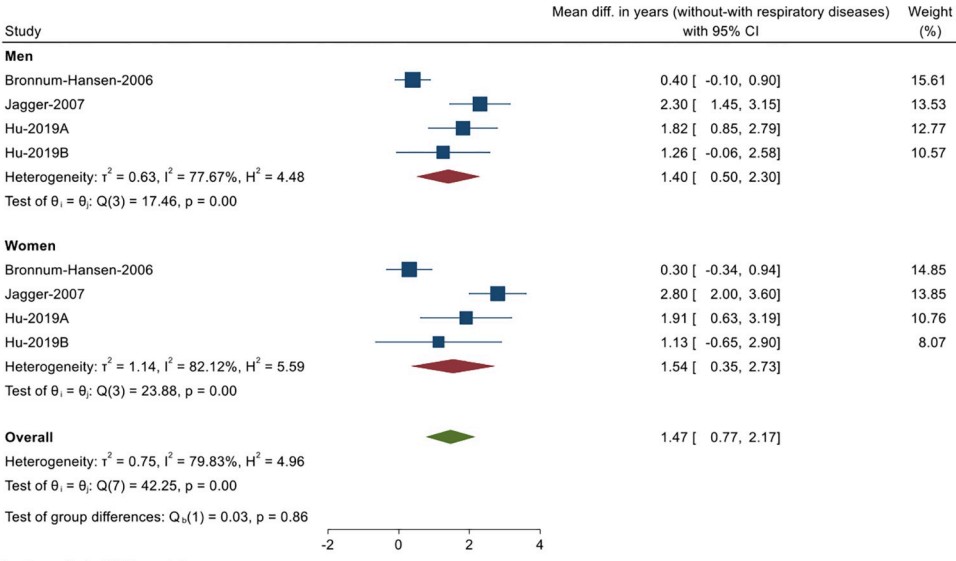

**Fig 3. Meta-analysis of mean difference in disability-free life expectancy (years) of individuals without respiratory diseases compared to those with respiratory diseases (based on data from 3 out of 9 studies reporting the impact of respiratory diseases on disability-free life expectancy; Hu et al 2019 reported estimates for two population samples, A = in 1990, B = in 2016).**

and DFLE (Table 10). Hearing loss/impairment had a small impact on LE and a greater impact on DFLE reduction [10,11]. A similar pattern was observed for visual impairment although the impact was slightly greater than that of hearing impairment [10,11]. Five studies examined the impact of dementia [14,33,35] or cognitive impairment [10,13] (Table 11). These conditions generally resulted in shorter LE, generally greater for women than men, but with greater effects on DFLE. Four studies [7,8,13,37] examined the impact of depression on LE and DFLE (Table 12). Depression and emotional problems had a greater impact on reduction of DFLE (or HALE in one study [37]) than LE although differences in DFLE between men and women were not consistent across studies (see S2 Results for full details).

## Ranking of LTCs based on studies assessing multiple conditions

Sixteen of the included studies assessed multiple LTCs for ages between 55 and 70 in different clusters. As cross-sectional studies that derived LTCs from cause of death data will underestimate the impact of non-fatal conditions such as arthritis, we report findings by type of study.

Of the nine cross-sectional studies assessing the impact of multiple LTCs, six found the greatest impact on DFLE was for elimination of heart disease and other circulatory diseases [11,12,16–18,29]. Diabetes and, to a lesser extent, cancer featured among the top conditions that would generate gains in DFLE or in the proportion of years lived free of disability if eliminated [12,14,16–18,27]. Of the seven longitudinal studies assessing the impact of multiple LTCs, stroke followed by diabetes were the conditions with the greatest impact on LE loss in the majority of studies assessing those LTCs [6,10,15,38] (stroke, range: 2.7 to 15.1 years; diabetes, range: 4.0 to 13.5 years). Three found the greatest effect on DFLE was also attributed to stroke for both genders.

We identified six studies reporting global YLDs estimates [42–47] and one study reporting HALE at birth [48] for multiple relevant LTCs. Nine LTCs included in the systematic review also featured among the most common causes of global YLDs (ranking was not available for

**Table 4. Summary data for the impact of cardiovascular diseases on life expectancy and disability-free life expectancy (or health-adjusted life expectancy) for men and women.**

| Study (year) | Survey | At age | LTC measured | Difference in LE Mean (95% CI) | | Difference in DFLE (or HALE) Mean (95% CI) | |
|---|---|---|---|---|---|---|---|
| | | | | Men | Women | Men | Women |
| *Laditka (2016) [13] | Panel Study of Income Dynamics (USA) | 55 years | IHD (vs. no IHD) | AA: 3.7 W: 5.8 | AA: 6.1 W: 11.5 | AA: 9.1 W: 10.0 | AA: 11.0 W: 15.7 |
| *Campolina (2013) [16] | SABE study (Brazil) | 60 years | IHD (vs. no IHD) | Gain†: 7.9 | Gain: 2.7 | Gain: 7.7 | Gain: 8.6 |
| *Campolina (2014) [17] | SABE study (Brazil) | 60 years | IHD (vs. no IHD) | Gain: 3.4 | Gain: 2.2 | Gain: 6.7 | Gain: 11.5 |
| *Bronnum-Hansen (2006) [12] | Danish Health Interview Survey 2000 | 65 years | IHD (vs. no IHD) | Gain: 1.7 | Gain: 1.4 | Gain: 1.2 | Gain: 0.8 |
| Diehr (1998) [30] | Cardiovascular Health Study (USA) | 65 years | CVD (vs. no CVD) | .. | | 1.0 | |
| *Hu (2019) [29] | Global Burden of Disease Study 2016 for China | 65 years | CVD (vs. no CVD) | .. | .. | Cohort 1: 3.2 (2.2 to 4.2) Cohort 2: 5.9 (4.4 to 7.3) | Cohort 1: 4.1 (2.7 to 5.4) Cohort 2: 7.2 (5.2 to 9.2) |
| *Jagger (2007) [41] | MRC Cognitive Function and Ageing Study (UK) | 65 years | Coronary heart disease (vs. no LTC) | 3.4 (2.7 to 4.1) | 2.8 (2.0 to 3.5) | 3.1 (2.3 to 3.8) | 3.3 (2.5 to 4.1) |
| | | | Peripheral vascular disease (vs. no LTC) | 2.8 (1.6 to 4.0) | 3.4 (2.0 to 4.8) | 2.8 (1.4 to 4.1) | 2.9 (1.5 to 4.3) |
| *Mathers (1999) [11] | Survey of disability, ageing and carers (Australia) | 65 years | IHD (vs. no IHD) | Gain: 2.6 | Gain: 1.1 | Gain: 1.9 | Gain: 0.9 |
| *Murtaugh (2011) [14] | National Mortality Followback Survey (USA) | 65 years | Heart attack | 1.0 | 0.3 | DLE: 0.1 | DLE: 0.0 |
| *Nusselder (1996) [18] | Dutch National Survey of General Practice | 65 years | IHD (vs. no IHD) | Gain: 3.1 | Gain: 2.7 | Gain: 1.5 | Gain: 0.9 |
| *Hayward (1998) [40] | Longitudinal Study of Aging (USA) | 70 years | Heart disease (vs. no LTC) | Gain: 3.1 | Gain: 3.9 | Gain: 1.9 | Gain: 1.5 |
| *Reynolds (2008b) [8] | Asset and Health Dynamics Among the Oldest Old-AHEAD (USA) | 70 years | Heart disease (vs. no LTC) | 3.2 (2.1 to 4.3) | 4.2 (3.2 to 5.2) | 2.9 (2.0 to 3.8) | 3.9 (3.1 to 4.7) |

*study measured multiple LTCs

†: Gain in LE/DFLE/HALE after elimination of LTC.

Difference in LE/DFLE: Years without LTC minus years with LTC, AA: African American, CVD: Cardiovascular disease, DFLE: Disability-free life expectancy, DLE: Life expectancy with disability, HALE: Health-adjusted life expectancy, IHD: Ischaemic heart disease, LTC: Long-term condition, W: White.

all LTCs included in the review). Studies generally reported YLDs in thousands, for all ages, and both sexes combined except one study that reported estimates for men and women separately [42]. Although the Global Burden of Disease (GBD) studies have used slightly different methods to calculate YLDs across the years, S3 Fig shows the ranking of these nine LTCs from 1990 to 2019 for comparison (lower rankings indicate greater disability attributed to the condition) and S4 Fig the number of YLDs in thousands for each relevant LTC. Depressive disorders were reported in five studies [42,44–47] and remained among the top five causes of global YLDs from 1990 to 2017. Diabetes has been consistently among the top ten causes of YLDs globally showing an increasing trend in the rankings (higher YLDs) in the same studies, followed by COPD which has been ranked between 5th and 14th cause of YLDs between 1990 and 2017 [42,44–47]. Hearing loss was among the top causes in two studies [46,47] ranging from 5th to 13th most common cause of YLDs. Increasing global trends in YLDs were also observed for osteoarthritis in the last two decades ranging from 17th place in 1990 to 11th in 2010 and

**Table 5. Summary data for the impact of hypertension on life expectancy and disability-free life expectancy (or health-adjusted life expectancy) for men and women.**

| Study (year) | Survey | At age | LTC measured | Difference in LE Mean (95% CI) | | Difference in DFLE (or HALE) Mean (95% CI) | |
|---|---|---|---|---|---|---|---|
| | | | | Men | Women | Men | Women |
| *Laditka (2016) [13] | Panel Study of Income Dynamics (USA) | 55 years | Hypertension (vs. no LTC) | AA: 4.5 W: 6.2 | AA: 7.3 W: 13.5 | AA: 9.5 W: 10.1 | AA: 9.9 W: 16.9 |
| *Public Health Agency of Canada report (2012) [27] | Canadian Community Health Survey | 55 years | Hypertension (vs. no LTC) | 2.1 | 1.5 | 2.7 | 2.0 |
| *Campolina (2013) [16] | SABE study (Brazil) | 60 years | Hypertension (vs. no LTC) | Gain†: 6.95 | Gain: 2.15 | Gain: 6.71 | Gain: 7.90 |
| *Campolina (2014) [17] | SABE study (Brazil) | 60 years | Hypertension (vs. no LTC) | Gain: 2.72 | Gain: 1.85 | Gain: 5.92 | Gain: 11.0 |
| * Liang (2020) [15] | Taiwan Longitudinal Study on Aging | 65 years | Hypertension (vs. no LTC) | -0.77 | -0.23 | -0.25 | 0.49 |
| *Mathers (1999) [11] | Survey of disability, ageing and carers (Australia) | 65 years | Hypertensive disease (vs. no LTC) | Gain: 0.06 | Gain: 0.04 | Gain in HALE: 0.13 | Gain in HALE: 0.15 |

*study measured multiple LTCs

†: Gain in LE/DFLE/HALE after elimination of LTC.

Difference in LE/DFLE: Years without LTC minus years with LTC, AA: African American, DFLE: Disability-free life expectancy, HALE: Health-adjusted life expectancy, LTC: Long-term condition, W: White.

12[th] in 2016 [42,44–47]. Stroke [42,44] and other cerebrovascular diseases [45] were reported in three studies and have also been associated with increased YLDs in recent years. Dementia [45–47] and ischaemic heart disease [44,45,47] were among the leading causes of YLDs in three studies shifting from rankings near the 30[th] places to the mid-20s in the period between 1990 and 2016, again indicating increased YLDs for those conditions. YLDs for cardiovascular diseases doubled from 17.7 million (CI: 12.9 to 22.5 million) to 34.4 million (CI: 24.9 to 43.6 million) over the period 1990–2019 [43].

**Table 6. Summary data for the impact of cerebrovascular disease on life expectancy and disability-free life expectancy (or health-adjusted life expectancy) for men and women.**

| Study (year) | Survey | At age | LTC measured | Gain in LE Mean (95% CI) | | Gain in DFLE (or HALE) Mean (95% CI) | |
|---|---|---|---|---|---|---|---|
| | | | | Men | Women | Men | Women |
| *Campolina (2013) [16] | SABE study (Brazil) | 60 years | Cerebrovascular disease (vs. no LTC) | 7.33 | 2.44 | 6.47 | 6.83 |
| *Campolina (2014) [17] | SABE study (Brazil) | 60 years | Cerebrovascular disease (vs. no LTC) | 3.03 | 2.04 | 5.7 | 9.73 |
| *Chen (2014) [39] | The 2006 China Disability Survey | 60 years | Cerebrovascular disease (vs. no LTC) | .. | .. | DLE: 0.45 | DLE: 0.39 |
| *Bronnum-Hansen (2006) [12] | Danish Health Interview Survey 2000 | 65 years | Cerebrovascular disease (vs. no LTC) | 0.6 | 0.8 | 0.7 | 0.6 |
| *Mathers (1999) [11] | Survey of disability, ageing and carers (Australia) | 65 years | Cerebrovascular disease (vs. no LTC) | 0.72 | 0.49 | 0.65 | 0.57 |
| *Hayward (1998) [40] | Longitudinal Study of Aging (USA) | 70 years | Cerebrovascular disease (vs. no LTC) | 0.4 | 0.6 | 0.3 | 0.3 |

*study measured multiple LTCs, †: Gain in LE/DFLE/HALE after elimination of LTC.

DFLE: Disability-free life expectancy, DLE: Life expectancy with disability, HALE: Health-adjusted life expectancy, LTC: Long-term condition.

**Table 7. Summary data for the impact of stroke on life expectancy and disability-free life expectancy (or health-adjusted life expectancy) for men and women.**

| Study (year) | Survey | At age | LTC measured | Difference in LE Mean (95% CI) | | Difference in DFLE (or HALE) Mean (95% CI) | |
|---|---|---|---|---|---|---|---|
| | | | | Men | Women | Men | Women |
| *Laditka (2016) [13] | Panel Study of Income Dynamics (USA) | 55 years | Stroke (vs. no stroke) | AA: 2.7 W: 5.6 | AA: 9.7 W: 15.1 | AA: 12.1 W: 14.8 | AA: 20.1 W: 24.2 |
| Chiu (2019) [32] | Nihon University Japanese Longitudinal Study of Aging | 65 years | Stroke (vs. no stroke) | 3.1 (2.0 to 4.6) | 3.2 (0.9 to 5.2) | 5.2 (4.0 to 6.8) | 5.0 (3.1 to 6.6) |
| Fang (2009) [28] | Beijing Multidimensional Longitudinal study on Aging (China) | 65 years | Stroke (vs. no stroke) | Cohort 1: 4.1 Cohort 2: 2.5 | | Cohort 1: 4.8 Cohort 2: 5.4 | |
| *Jagger (2007) [41] | MRC Cognitive Function and Ageing Study (UK) | 65 years | Stroke (vs. no stroke) | Gain†: 4.8 (3.8 to 5.8) | Gain: 4.6 (3.5 to 5.7) | Gain: 6.5 (5.4 to 7.7) | Gain: 5.8 (4.5 to 7.1) |
| *Murtaugh (2011) [14] | National Mortality Followback Survey (USA) | 65 years | Stroke (vs. no stroke) | -1.1 | -1.5 | DLE: 0.4 | DLE: 0.5 |
| *Reynolds (2008b) [8] | Asset and Health Dynamics Among the Oldest Old-AHEAD (USA) | 70 years | Stroke (vs. no stroke) | 6.2 | 6.2 | 6.6 | 6.5 |

*study measured multiple LTCs

†: Gain in LE/DFLE/HALE after elimination of LTC.

Difference in LE/DFLE: Years without LTC minus years with LTC, AA: African American, DFLE: Disability-free life expectancy, DLE: Life expectancy with disability, HALE: Health-adjusted life expectancy, LTC: Long-term condition, W: White.

**Table 8. Summary data for the impact of cancer on life expectancy and disability-free life expectancy (or health-adjusted life expectancy) for men and women.**

| Study (year) | Survey | At age | LTC measured | Difference in LE Mean (95% CI) | | Difference in DFLE (or HALE) Mean (95% CI) | |
|---|---|---|---|---|---|---|---|
| | | | | Men | Women | Men | Women |
| *Belanger (2002) [38] | National Population Health Survey (Canada) | 45 years | Cancer (vs. no cancer) | 11.0 | 12.9 | 6.4 | 5.3 |
| *Campolina (2013) [16] | SABE study (Brazil) | 60 years | Cancer (vs. no cancer) | Gain†: 8.12 | Gain: 3.04 | Gain: 7.32 | Gain: 8.46 |
| *Campolina (2014) [17] | SABE study (Brazil) | 60 years | Cancer (vs. no cancer) | Gain: 3.85 | Gain: 2.77 | Gain: 6.54 | Gain: 11.57 |
| *Bronnum-Hansen (2006) [12] | Danish Health Interview Survey 2000 | 65 years | Neoplasms (vs. no neoplasms) | Gain: 2.3 | Gain: 2.2 | Gain: 1.5 (0.93 to 2.07) | Gain: 1.3 (0.63 to 1.97) |
| *Hu (2019) [29] | Global Burden of Disease Study 2016 for China | 65 years | Cancer (vs. no cancer) | .. | .. | Cohort 1: 1.18 (0.21 to 2.12) Cohort 2: 1.84 (0.47 to 3.21) | Cohort 1: 0.86 (-0.38 to 2.10) Cohort 2: 1.09 (-0.71 to 2.89) |
| *Mathers (1999) [11] | Survey of disability, ageing and carers (Australia) | 65 years | Neoplasms (vs. no neoplasms) | Gain in LE: 2.40 | Gain in LE: 1.07 | Gain in HALE: 1.83 | Gain in HALE: 0.91 |
| *Nusselder (1996) [18] | Dutch National Survey of General Practice | 65 years | Cancer (vs. no cancer) | Gain: 2.7 | Gain: 1.9 | Gain: 0.9 | Gain: 0.4 |
| *Public Health Agency of Canada report (2012) [27] | Canadian Community Health Survey | 65 years | Cancer (vs. no cancer) | 11.5 (11.4 to 11.6) | 13.7 (13.6 to 13.8) | 9.2 (8.7 to 9.7) | 10.3 (9.9 to 10.7) |
| *Hayward (1998) [40] | Longitudinal Study of Aging (USA) | 70 years | Cancer (vs. no cancer) | 1.6 | 1.2 | Gain: 1.2 | Gain: 0.6 |
| *Reynolds (2008b) [8] | Asset and Health Dynamics Among the Oldest Old-AHEAD (USA) | 70 years | Cancer (vs. no cancer) | 4.2 (2.9 to 5.5) | 3.8 (2.5 to 5.2) | 3.8 (2.7 to 4.9) | 2.7 (1.7 to 3.7) |

*study measured multiple LTCs

†: Gain in LE/DFLE/HALE after elimination of LTC.

Difference in LE/DFLE: Years without LTC minus years with LTC, DFLE: Disability-free life expectancy, HALE: Health-adjusted life expectancy, LTC: Long-term condition.

**Table 9. Summary data for the impact of arthritis on life expectancy and disability-free life expectancy (or health-adjusted life expectancy) for men and women.**

| Study (year) | Survey | At age | LTC measured | Difference in LE Mean (95% CI) | | Difference in DFLE (or HALE) Mean (95% CI) | |
|---|---|---|---|---|---|---|---|
| | | | | Men | Women | Men | Women |
| *Belanger (2002) [38] | National Population Health Survey (Canada) | 45 years | Arthritis (vs. no arthritis) | 2.2 | 3.3 | 6.5 | 8.8 |
| *Laditka (2016) [13] | Panel Study of Income Dynamics (USA) | 55 years | Arthritis (vs. no arthritis) | AA: 0.4 W: 7.3 | AA: 4.8 W: 9.6 | AA: 6.2 W: 11.7 | AA: 11.5 W: 15.2 |
| *Murtaugh (2011) [14] | National Mortality Followback Survey (USA) | 65 years | Arthritis (vs. no arthritis) | -0.5 | -0.1 | DLE: 0.3 | DLE: 0.4 |
| *Jagger (2007) [41] | MRC Cognitive Function and Ageing Study (UK) | 65 years | Arthritis (vs. no arthritis) | Gain†: 0.2 (-0.5 to 0.8) | Gain: -0.2 (-0.7 to 0.4) | Gain: 1.0 (0.3 to 1.7) | Gain: 2.6 (1.9 to 3.3) |
| Reynolds (2008a) [6] | Asset and Health Dynamics Among the Oldest Old-AHEAD (USA) | 70 years | Arthritis (vs. no arthritis) | 0.4 (-0.76 to 1.56) | 0.6 (-0.44 to 1.64) | 1.7 (0.76 to 2.64) | 2.7 (1.95 to 3.45) |
| *Chen (2014) [39] | The 2006 China Disability Survey | 60 years | Osteoarthritis (vs. no osteoarthritis) | .. | .. | DLE with LTC: 0.27 | DLE with LTC: 0.48 |
| *Mathers (1999) [11] | Survey of disability, ageing and carers (Australia) | 65 years | Osteoarthritis (vs. no osteoarthritis) | Gain: 0.01 | Gain: 0.0 | Gain in HALE: 0.28 | Gain in HALE: 0.85 |
| | | | Rheumatoid arthritis (vs. no rh. arthritis) | .. | Gain: 0.01 | .. | Gain in HALE: 0.11 |

*study measured multiple LTCs

†: Gain in LE/DFLE/HALE after elimination of LTC.

Difference in LE/DFLE: Years without LTC minus years with LTC, AA: African American, ADL: Activities of daily living, DFLE: Disability-free life expectancy, DLE: Life expectancy with disability, HALE: Health-adjusted life expectancy, LTC: Long-term condition, W: White.

**Table 10. Summary data for the impact of sensory loss on life expectancy and disability-free life expectancy (or health-adjusted life expectancy) for men and women.**

| Study (year) | Survey | At age | LTC measured | Difference in LE Mean (95% CI) | | Difference in DFLE (or HALE) Mean (95% CI) | |
|---|---|---|---|---|---|---|---|
| | | | | Men | Women | Men | Women |
| *Chen (2014) [39] | The 2006 China Disability Survey | 60 years | Hearing loss (vs. no loss) | .. | .. | LED with LTC: 1.58 | LED with LTC: 1.55 |
| *Mathers (1999) [11] | Survey of disability, ageing and carers (Australia) | 65 years | Hearing loss (vs. no loss) | 0.0 | 0.0 | Gain† in HALE: 0.14 | Gain in HALE: 0.16 |
| *Jagger (2007) [41] | MRC Cognitive Function and Ageing Study (UK) | 65 years | Hearing impairment (vs. no impairment) | Gain: 0.3 (-0.5 to 1.0) | Gain: 0.2 (-0.6 to 0.9) | Gain: 0.5 (-0.3 to 1.3) | Gain: 1.2 (0.3 to 2.1) |
| | | | Visual impairment (vs. no impairment) | Gain: 0.9 (0.0 to 1.9) | Gain: 1.5 (0.7 to 2.3) | Gain: 2.0 (0.9 to 3.1) | Gain: 3.1 (2.1 to 4.0) |
| Tareque (2019) [9] | Panel on Health and Ageing in Singaporean Elderly | 60 years | Hearing impairment (vs. no impairment) | 0.5 (-3.1 to 5.1) | -0.4 (-4.5 to 2.6) | -2.1 (-5.0 to 1.7) | 1.0 (-2.0 to 3.6) |
| | | | Visual impairment (vs. no impairment) | -2.0 (-4.7 to 1.4) | -2.6 (-5.5 to 1.9) | -4.2 (-6.8 to -0.7) | -4.4 (-7.3 to -1.8) |
| | | | Hearing & visual impairment (vs. neither) | -3.9 (-6.7 to -1.1) | -3.4 (-6.7 to -0.6) | -8.0 (-10.5 to -4.9) | -5.4 (-9.1 to -2.8) |

*study measured multiple LTCs

†: Gain in LE/DFLE/HALE after elimination of LTC.

Difference in LE/DFLE: Years without LTC minus years with LTC, DFLE: Disability-free life expectancy, HALE: Health-adjusted life expectancy, LED: Life expectancy with disability, LTC: Long-term condition.

**Table 11. Summary data for the impact of dementia and cognitive impairment on life expectancy and disability-free life expectancy (or health-adjusted life expectancy) for men and women.**

| Study (year) | Survey | At age | LTC measured | Difference in LE Mean (95% CI) | | Difference in DFLE (or HALE) Mean (95% CI) | |
|---|---|---|---|---|---|---|---|
| | | | | Men | Women | Men | Women |
| **Manton (1991) [35]** | National Long Term Care Surveys (USA) | 65 years | Dementia (vs. no dementia) | .. | .. | Gain†: 0.22 | Gain: 0.48 |
| **Murtaugh (2011) [14]** | National Mortality Followback Survey (USA) | 65 years | Dementia (vs. no dementia) | -3.3 | -4.4 | DLE: 2.2 | DLE: 1.6 |
| **Dodge (2003) [33]** | MoVIES survey (USA) | 70 years | Alzheimer's disease (vs. no disease) | 6.9 | 6.7 | 7.1 | 8.5 |
| **Laditka (2016) [13]** | Panel Study of Income Dynamics (USA) | 55 years | Memory impairment (vs. no impairment) | AA: 1.5 W: 2.4 | AA: 4.6 W: 8.3 | AA: 8.6 W: 7.0 | AA: 9.0 W: 14.0 |
| **Jagger (2007) [41]** | MRC Cognitive Function and Ageing Study (UK) | 65 years | Cognitive impairment (vs. no impairment) | Gain: 3.4 (2.3–4.6) | Gain: 3.6 (2.7–4.6) | Gain: 4.2 (2.6–5.8) | Gain: 4.4 (2.9–5.8) |

*study measured multiple LTCs

†: Gain in LE/DFLE/HALE after elimination of LTC.

Difference in LE/DFLE: Years without LTC minus years with LTC, AA: African American, DFLE: Disability-free life expectancy, DLE: Life expectancy with disability, HALE: Health-adjusted life expectancy, LTC: Long-term condition, W: White.

## Discussion

This systematic review provides the first comprehensive evidence synthesis of the effect of a range of LTCs on disability-free or healthy, and total life expectancy. For two LTCs meta-analyses could be performed, resulting in estimated gains of 2 years disability-free at age 65 for those without compared to those with diabetes (pooled mean difference in DFLE from four studies = 2.28 years, 95% CI: 0.57–3.99, p<0.01, I$^2$ = 96.7%), and gains of 1.5 years disability-

**Table 12. Summary data for the impact of depression on life expectancy and disability-free life expectancy (or health-adjusted life expectancy) for men and women.**

| Study (year) | Survey | At age | LTC measured | Difference in LE Mean (95% CI) | | Difference in DFLE (or HALE) Mean (95% CI) | |
|---|---|---|---|---|---|---|---|
| | | | | Men | Women | Men | Women |
| ***Laditka (2016) [13]** | Panel Study of Income Dynamics (USA) | 55 years | Depression (vs. no depression) | AA: 3.3 W: 1.9 | AA: 2.8 W: 7.3 | AA: 8.9 W: 5.7 | AA: 8.9 W: 12.4 |
| **Pérès (2008) [7]** | MRC Cognitive Function and Ageing Study (UK) | 65 years | Depression (vs. no depression) | 3.4 (-0.7 to 7.5) | 0.8 (-2.0 to 3.6) | 3.7 (0.2 to 7.2) | 1.4 (-2.1 to 4.9) |
| | | | Emotional problems (vs. no emotional problems) | 1.0 (-1.8 to 3.7) | 1.2 (-0.9 to 3.3) | 1.5 (-1.4 to 4.4) | 1.8 (-0.4 to 4.0) |
| **Steensma (2016) [37]** | Canadian Community Health Survey National Population Health Survey for mortality rates | 65–69 years | Depression (vs. no depression) | 2.1 (0.1 to 4.1) | 3.2 (1.8 to 4.6) | 6.0 (3.8 to 8.2) | 6.7 (5.3 to 8.1) |
| ***Reynolds (2008b) [8]** | Asset and Health Dynamics Among the Oldest Old (USA) | 70 years | Depression without comorbidities (vs. no depression or comorbidities) | 3.5 (1.21 to 5.79) | 2.9 (1.11 to 4.69) | 6.5 (4.77 to 8.23) | 4.2 (2.92 to 5.47) |

*study measured multiple LTCs.

Difference in LE/DFLE: Years without LTC minus years with LTC, AA: African American, DFLE: Disability-free life expectancy, HALE: Health-adjusted life expectancy, LTC: Long-term condition, W: White.

free at age 65 for those without compared to those with respiratory disease (pooled mean difference in DFLE = 1.47 years, 95% CI: 0.77–2.17, p<0.01, $I^2$ = 79.8%). Narrative synthesis of remaining studies suggested that many LTCs have a greater effect on DFLE/HALE than LE, suggesting that elimination of certain conditions including stroke, diabetes, hypertension, and arthritis may result in compression of disability. Evidence for the remaining conditions (e.g., respiratory, cancer, dementia) is mixed.

Diabetes is known for multiple vascular and neuropathic complications and increased risk of disability including difficulties with the ability to carry out ADLs and loss of mobility [49]. Findings from longitudinal data included in this review, and taking into account incidence and recovery from disability, also show that individuals with diabetes have an earlier onset of disability compared to those without the condition, and a lower probability of recovering from functional limitations [4,31]. Although there is some evidence from US cohorts for gains in DFLE in people with diabetes in the past 20 years [5], included studies highlight that diabetes still has a substantial impact on disability-free years. Estimates from the Global Burden of Disease Study [44] also place diabetes among the top ten leading causes of increased years with disability in recent decades. With population ageing, the prevalence of diabetes will also increase and so will the need for disability-related health resources.

Although not all of the included studies assessed all the LTCs considered in the systematic review, data from sixteen studies assessing multiple LTCs allowed a crude ranking of these conditions in terms of their impact on DFLE and LE. Stroke had the strongest impact on DFLE (range in years: 4.8–24.2) and it was greater than that on LE, especially when individuals were initially classified as ADL/IADL disabled. Similarly, hypertension and cardiovascular disease were among the LTCs with the greatest effect on DFLE and LE with most studies signifying a compression of disability after disease elimination. Arthritis was the only non-fatal condition within the top five LTCs affecting DFLE more than LE. Arthritis was associated with a small loss of LE which was generally greater in women, and studies indicated that elimination of the condition would result in one of the greatest gains in DFLE/HALE. Although not directly comparable due to different estimation methods, the identified DALY-related studies also show increasing trends in global YLDs for stroke, cardiovascular diseases, and osteoarthritis which further supports the high disability burden linked to these LTCs and the potential gains from their prevention and improved management.

Evidence suggests that multimorbidity predicts future functional decline, with greater decline in people with higher number of LTCs and greater disease severity [50]. The few studies that assessed the effect of comorbidities in this review show a rather complex picture where multiple LTCs appear to generally reduce DFLE and occasionally LE compared to those without the conditions, but the effect is not necessarily additive and seems to vary by the combination of LTCs studied. In recent years, there has been a movement toward research of clusters of chronic conditions and implementation of services based on comorbidity/multimorbidity. However, there is limited evidence from longitudinal data on the disease combinations that are more or less disabling, especially in terms of DFLE, despite projections indicating that complex multimorbidity (four or more diseases) will double in the next 15 years and gains in LE will be spent mostly with complex multimorbidity [2]. Understanding how multimorbidity combinations relate to disabled and disability-free life expectancy is therefore an important step towards planning for appropriate future health and social care provision and designing interventions.

This review has several strengths and limitations. We performed comprehensive search strategies to identify published research including major electronic databases and reference list checking. Along with synthesising evidence narratively across the LTCs, we conducted meta-analyses for a subgroup of studies. However, this was feasible only for two LTCs (diabetes,

respiratory diseases) and only for DFLE as the outcome. We also note it is likely the presented Egger tests and funnel plots were underpowered to detect small-study effects bias, while the high heterogeneity observed could not be further explored by subgroup analysis or meta-regressions given the limited number of included studies. Underreporting of HE estimates within the primary studies, particularly standard errors or 95% confidence intervals for LE and DFLE estimates was a major limitation that prevented further quantitative analyses for most of the LTCs. Other methodological differences such as the small number of studies for some LTCs, different outcomes (DFLE versus HALE) reported at different ages (e.g., 55, 60, 65) also meant that many studies did not meet the predefined criteria for meta-analyses. We assessed the quality of the studies, but we had to supplement a standard tool since no assessment tool is available for HE studies. LTCs were mostly self-reported, which means that people with undiagnosed conditions would be classified as not having the LTC, thereby potentially underestimating the difference in DFLE between people with and without the LTC. Guidelines for the identification and management of conditions, such as diabetes and hypertension, have also changed over the decades covered in the included studies which may have also biased DFLE and LE differences between those with and without the LTC. Disability measurement was also largely based on self-report and agreement with objective measurements may vary by LTC status.

Most of the LTCs with the greatest effect on DFLE reviewed here are strongly associated with unhealthy lifestyles. Early interventions to reduce known risk factors such as smoking, physical inactivity and obesity in younger adults could prevent, delay or significantly reduce disability and allow individuals to live independently with minimal or mild disability in older age. Healthier lifestyles including healthy dietary patterns have been associated with reduced risk of many chronic conditions including diabetes, cardiovascular disease and dementia, and there are also findings supporting reduced risk of developing self-reported disability [51]. An intensive lifestyle intervention targeting weight loss and improved fitness was related to a 50% slower decline in physical disability among overweight adults with diabetes and an increase in number of disability-free years compared to a group receiving diabetes support and education [52,53]. Several studies have also shown that pain and disability improve with short-term exercise programmes in patients with osteoarthritis [54]. Many older adults perceive health as functional capability rather than physical fitness, with the ability to master daily life as a vital component [55]. Therefore, improving the number of disability-free years over time is as important as focusing on the prevention and management of LTCs, to allow older adults to be able to do the things to which they attribute value. A recent review [56] indicated that changing both personal and contextual factors can help older adults engage in ADL and IADL. Interventions that included exercise, cognitive behavioural therapy, problem-solving and environmental modifications as the main components were likely to be more effective at reducing disability [56].

Our novel evidence synthesis of the impact of LTCs on DFLE at older ages has identified a number of LTCs that, if eliminated, have the potential to make substantial gains in DFLE. Further studies are needed to provide stronger evidence for many of the LTCs considered, as well as combinations of LTCs to assess specific multiple conditions. However, all further studies should ensure reporting of the uncertainty around estimates of the gain in both DFLE/HALE and LE, to enable further meta-analyses.

## Supporting information

**S1 Checklist.**
(DOCX)

**S1 Fig. Funnel plot for studies examining the impact of diabetes on DFLE in men and women.**
(TIF)

**S2 Fig. Funnel plot for studies examining the impact of respiratory diseases on DFLE in men and women.**
(TIF)

**S3 Fig. Ranking of most common causes of global YLDs corresponding to the LTCs assessed within the systematic review.**
(TIF)

**S4 Fig. Global years lived with disability (YLDs) for 14 LTCs included in the systematic review.**
(TIF)

**S1 Table. Quality assessment for 13 cross-sectional studies.**
(DOCX)

**S2 Table. Quality assessment for 17 longitudinal studies.**
(DOCX)

**S1 Methods. Key concepts.**
(DOCX)

**S2 Methods. Search strategy example.**
(DOCX)

**S3 Methods. Quality assessment.**
(DOCX)

**S1 Results. Quality assessment.**
(DOCX)

**S2 Results. Effects of LTCs on health expectancy (narrative syntheses).**
(DOCX)

**S1 References.**
(DOCX)

## Author Contributions

**Conceptualization:** Andrew Kingston, Carol Jagger.

**Data curation:** Ilianna Lourida, Holly Q. Bennett, Fiona Beyer, Andrew Kingston, Carol Jagger.

**Formal analysis:** Ilianna Lourida, Holly Q. Bennett, Andrew Kingston, Carol Jagger.

**Funding acquisition:** Andrew Kingston.

**Project administration:** Andrew Kingston.

**Supervision:** Andrew Kingston, Carol Jagger.

**Writing – original draft:** Ilianna Lourida, Holly Q. Bennett, Andrew Kingston, Carol Jagger.

**Writing – review & editing:** Ilianna Lourida, Holly Q. Bennett, Fiona Beyer, Andrew Kingston, Carol Jagger.

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
