## [Decision Letter · Decision Letter 0]

8 Feb 2022

PGPH-D-21-01155

The impact of long-term conditions on disability-free life expectancy: a systematic review

Dear Dr. Kingston,

Thank you for submitting your manuscript to PLOS Global Public Health. After careful consideration, we feel that it has merit but does not fully meet PLOS Global Public Health’s publication criteria as it currently stands. Therefore, we invite you to submit a revised version of the manuscript that addresses the points raised during the review process.

We look forward to receiving your revised manuscript.

Kind regards,

Guglielmo Campus, Ph.D DDS

Academic Editor

Journal Requirements:

1. We ask that a manuscript source file is provided at Revision. Please upload your manuscript file as a .doc, .docx, .rtf or .tex. If you are providing a .tex file, please upload it under the item type ‘LaTeX Source File’ and leave your .pdf version as the item type ‘Manuscript’.

2. Please provide  separate figure files in .tif or .eps format only and remove any figures embedded in your manuscript file.  Please ensure that all files are under our size limit of 20MB.  

For more information about how to convert your figure files please see our guidelines: Once you've converted your files to .tif or .eps, please also make sure that your figures meet our format requirements

3. Please update the completed 'Competing Interests' statement, including any COIs declared by your co-authors. If you have no competing interests to declare, please state "The authors have declared that no competing interests exist".

4. We notice that your supplementary [figures/tables] are included in the manuscript file.  Please remove them and upload them  with the file type 'Supporting Information'  . Please ensure that all Supporting Information files are included correctly and that each one has a legend listed in the manuscript after the references list. 

5. Please add a full list of legends for all supporting information files (including figures, table and data files) after the references list. 

6. Please provide a complete Data Availability Statement in the submission form, ensuring you include all necessary access information or a reason for why you are unable to make your data freely accessible. Note that it is not acceptable for the authors to be the sole named individuals responsible for ensuring data access.

PLOS defines a study's minimal data set as the underlying data used to reach the conclusions drawn in the manuscript and any additional data required to replicate the reported study findings in their entirety. Any potentially identifying patient information must be fully anonymized. 

If your research concerns only data provided within your submission, please write "All data are in the manuscript and/or supporting information files" as your Data Availability Statement.

Additional Editor Comments (if provided):

Reviewers' comments:

Reviewer's Responses to Questions

**Comments to the Author**

1. Does this manuscript meet PLOS Global Public Health’s publication criteria? Is the manuscript technically sound, and do the data support the conclusions? The manuscript must describe methodologically and ethically rigorous research with conclusions that are appropriately drawn based on the data presented.

Reviewer #1: Yes

Reviewer #2: Yes

2. Has the statistical analysis been performed appropriately and rigorously?

Reviewer #1: Yes

Reviewer #2: Yes

3. Have the authors made all data underlying the findings in their manuscript fully available (please refer to the Data Availability Statement at the start of the manuscript PDF file)?

Reviewer #1: Yes

Reviewer #2: Yes

4. Is the manuscript presented in an intelligible fashion and written in standard English?

Reviewer #1: Yes

Reviewer #2: No

5. Review Comments to the Author

Reviewer #1: Line number- 29, 132: Please, mention the name of month along with the year 2007.

Line number- 596, 625: Homogeneity should be maintained during writing references. As because all the references didn't contain doi number, you can omit the doi numbers.

Reviewer #2: Manuscript number PGPH-D-21-01155

Title The impact of long-term conditions on disability-free life expectancy: a systematic review

Overall comments: This manuscript focused on an important topic to estimate increases in DFLE associated with elimination of a range of long-term conditions (LTCs). They also made attempt to address the gap by conducting a systematic review of the literature to assess the effect of a range of LTCs, singly and in combination, on disability-free, healthy, and total life expectancy, and specifically, which LTCs effect on DFLE than LE. Overall this review address important issue but the current version required clarifications and minor revision.

Specific comments

• Is the term ‘long-term conditions’ (LTCs) used is standard or is it used for this current study purpose; if so definition to be given

• What is the PICO question for this review?

• It was very wage LTCs; it need to specifically mention that what diseases included in the review; time period (last ten or twenty years), countries (developed and developing); whether unpublished literature is included; other than English included or not

• What is the outcome parameter is not clear in the data extraction section

• Data extraction: Provide details of what data were extracted. What are the statistical tools used? any software used for search and analysis?

• If full paper not available what action was taken to get full paper for review; any missing information; how the missing information is managed; whether contacted author to provide information?

• How the quality of the evidence assessed?

• How risk of bias assessed; any standard check list used

• The statement “The funnel plot indicated potential publication bias. Results of the Egger test for small-study effects suggested this is unlikely to be problematic. Meta-analysis of LE estimates for the above four studies was not possible due to insufficient data” what authors want to convey from the results. If it is biased is it possible to remove or reduce bias and recalculate.

• Overall authors need to simplify or make it non-statistician to understand the results.

• Hu et al.2019 reported estimates for two population samples; is it from different region/ area; it seems to be same study for respiratory diseases also. It is calculated for age at 65 years; what about other age group.

• Ranking graphs needs explanation; it too much of information in one graph

Overall authors made good attempt to review, however the current version is not suitable for publication. I hope the comments are useful for authors to improve the article.

6. PLOS authors have the option to publish the peer review history of their article (what does this mean?). If published, this will include your full peer review and any attached files.

**Do you want your identity to be public for this peer review?** For information about this choice, including consent withdrawal, please see our Privacy Policy.

Reviewer #1: **Yes: **Dr. Afsana Mahjabin

MBBS, MPH (NIPSOM)

Assistant Professor

Department of Community Medicine

Monno Medical College

Manikganj, Bangladesh

Reviewer #2: **Yes: **Malaisamy Muniyandi

---

## [Decision Letter · Decision Letter 1]

14 Jun 2022

The impact of long-term conditions on disability-free life expectancy: a systematic review

PGPH-D-21-01155R1

Dear Dr. Kingston,

We are pleased to inform you that your manuscript 'The impact of long-term conditions on disability-free life expectancy: a systematic review' has been provisionally accepted for publication in PLOS Global Public Health.

Best regards,

Guglielmo Campus, Ph.D DDS

Academic Editor

I am delighted to accept this interesting paper.

Reviewer Comments (if any, and for reference):

Reviewer's Responses to Questions

**Comments to the Author**

1. If the authors have adequately addressed your comments raised in a previous round of review and you feel that this manuscript is now acceptable for publication, you may indicate that here to bypass the “Comments to the Author” section, enter your conflict of interest statement in the “Confidential to Editor” section, and submit your "Accept" recommendation.

Reviewer #3: All comments have been addressed

Reviewer #4: All comments have been addressed

2. Does this manuscript meet PLOS Global Public Health’s publication criteria? Is the manuscript technically sound, and do the data support the conclusions? The manuscript must describe methodologically and ethically rigorous research with conclusions that are appropriately drawn based on the data presented.

Reviewer #3: Yes

Reviewer #4: Yes

3. Has the statistical analysis been performed appropriately and rigorously?

Reviewer #3: I don't know

Reviewer #4: Yes

4. Have the authors made all data underlying the findings in their manuscript fully available (please refer to the Data Availability Statement at the start of the manuscript PDF file)?

Reviewer #3: Yes

Reviewer #4: Yes

5. Is the manuscript presented in an intelligible fashion and written in standard English?

Reviewer #3: Yes

Reviewer #4: Yes

6. Review Comments to the Author

Reviewer #3: None

Reviewer #4: • Congratulations to the research team because this is one of the first global, comprehensive systematic review and meta-analysis quantifying the impact of long-term conditions on health expectancy outcomes especially relating to stroke, hypertension, diabetes and arthritis. Their findings will help inform global health policy on the quality of life of patients with these conditions and the metrics of measurement of these health impacts.

• Small comments: Provide the data/reference for men in parenthesis in line 281 or remove the parenthesis

• The quality of the figures 1, 2, and 3 are poor. Is there a way the quality can be enhanced to make them more reader friendly and aesthetically attractive?

• A good attempt was made to respond to all previous comments by other reviewers.

7. PLOS authors have the option to publish the peer review history of their article (what does this mean?). If published, this will include your full peer review and any attached files.

**Do you want your identity to be public for this peer review?** For information about this choice, including consent withdrawal, please see our Privacy Policy.

Reviewer #3: No

Reviewer #4: No
